# Weakly supervised 3D ConvLSTMs for Monte-Carlo radiotherapy dose simulations

**Sonia Martinot** [1,2,3]                          SONIA.MARTINOT@CENTRALESUPELEC.FR
[1] *CentraleSupelec* [2] *Therapanacea* [3] *Institut Gustave Roussy*

**Norbert Bus**[2]                                        N.BUS@THERAPANACEA.EU
**Maria Vakalopoulou**[1]                   MARIA.VAKALOPOULOU@CENTRALESUPELEC.FR
**Charlotte Robert**[3]                             CH.ROBERT@GUSTAVEROUSSY.FR
**Eric Deutsch**[3]                                 ERIC.DEUTSCH@GUSTAVEROUSSY.FR
**Nikos Paragios** [2]                               N.PARAGIOS@THERAPANACEA.EU

## Abstract

Radiotherapy dose simulation using the Monte-Carlo technique surpasses existing algorithms in terms of precision but remains too time-consuming to be integrated in clinical workflows. We introduce a 3D recurrent and fully convolutional neural network architecture to produce high-precision Monte-Carlo-like dose simulations from low-precision and cheap-to-compute ones. We use the noise-to-noise setting, a weakly supervised training strategy, by training the models solely on low-precision data without expensive-to-compute, high-precision dose simulations. Several evaluation metrics are used to compare with other methods and to assess the clinical viability and quality of the generated dose maps. Code is available at https://git.io/JGai7.

**Keywords:** Deep Learning, Radiotherapy, Recurrent Neural Networks, Dose Simulations

## 1. Introduction

Delivering safe radiotherapy treatments requires accurate dose estimation prior to patient irradiation. Dose maps simulated with the Monte-Carlo (MC) technique remain the most precise as it models all particle-matter interactions. However, despite existing GPU accelerations for MC, simulating enough particles to compute a high-precision dose map with no inherent MC noise necessitates a prohibitive amount of computational resources which prevents its integration into clinical workflows. To overcome this limitation, prior work focusing on Intensity-Modulated Radiotherapy (IMRT) plans (Otto, 2008) trained models to denoise low-precision, few-particle MC dose simulations in order to generate clean dose maps similar to real high-precision MC dose simulations (Bai et al., 2020).

We propose a 3D, recurrent yet fully convolutional model based on ConvLSTMs (Shi et al., 2015). Able to process sequential data, the model should learn the causal relationship that links the number of simulated particles and the level of precision of the resulting dose map. Furthermore, considering a traditional supervised training strategy calls for a substantial dataset with high-precision and therefore expensive-to-compute ground-truth dose simulations. To circumvent this impeding requirement, we use the weakly supervised noise-to-noise (N2N) method (Lehtinen et al., 2018) such that the training set only needs to comprise cheap-to-compute noisy dose simulations.

Our contributions are two-fold: *(i)* we investigate progressive denoising of Volumetric Modulated Arc Therapy (VMAT) plans with a 3DConvLSTMs-based model, *(ii)* we train our model in a weakly supervised setting and compare it to BiONet (Xiang et al., 2020).

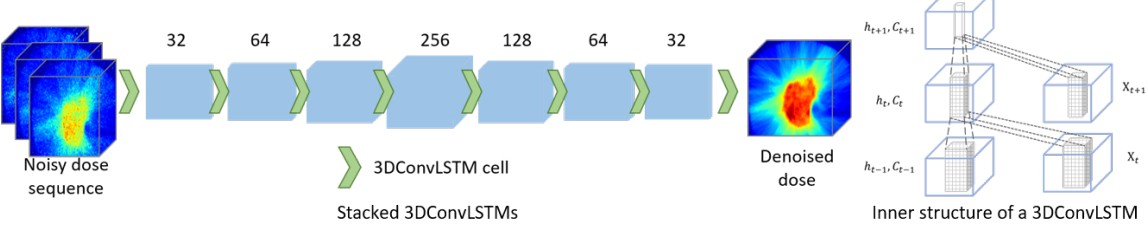

Figure 1: Left: Inner structure of a 3DConvLSTM cell. Right: our model's architecture with number of output channels indicated.

## 2. Material and Methods

The cohort comprises 50 VMAT patients. Treated anatomies include 22 pelvic and 28 head and neck cases. The cohort was split to 40, 5 and 5 patients for training, validation and test respectively, while anatomies were distributed evenly across sets. A model of the linear accelerator was implemented using OpenGate (Sarrut et al., 2014) to compute the corresponding dose simulation of each plan in the dataset. The resolution of the dose volumes was set to 2 $mm^3$. MC simulations were computed for $5e^8$, $1e^9$, $5e^9$ particles. For each patient, we computed 5 independent MC simulations with $5e^9$ particles for N2N training. Ground-truth dose maps used at evaluation time, were computed with $1e^{11}$ particles so the maximum uncertainty in areas within 20% - 100% of the dose maximum remained below the clinically accepted 3% threshold. Complete simulation of a ground-truth dose map required over 4k hours of computation time on CPU without using any variance reduction technique.

ConvLSTMs take advantage of their convolutional and recurrent nature to extract both spatial and sequential features from their input. We generalize the structure to cope with 3D data. As displayed in Figure 1, our model consists of 7 3DConvLSTM cells stacked on top of each other, without introducing any spatial down-sampling. All convolutions in the model have 3×3×3 filters. During N2N training, the input of the model is a patient's sequence of decreasingly noisy MC dose simulations ($5e^8$, $1e^9$ and $5e^9$ particles), while the reference output is another distinct noisy simulation computed with $5e^9$ particles for the same patient. For all considered architectures, training was patch-based with a patch size of 32x32x32 i.e. 6.4 $mm^3$ and a batch size of 8 for $1e^5$ iterations until difference in validation loss was inferior to $1e^{-2}$ for more than 1000 iterations using AdamW optimizer (learning rate=0.001, betas=(0.9, 0.999), weight decay=0.01). The loss function combines the Structural Similarity Index Measure (SSIM) and the L1 loss as follows: $Loss = 20 \times L1 + SSIM$.

## 3. Results

For benchmark, we generalized BiONet to handle 3D inputs. Table 1 shows the SSIM, the L1 loss and the clinically used Gamma Passing Rate (GPR) between the models' denoised volume and the corresponding clean, ground-truth dose volume ($1e^{11}$ particles) computed over the test set. The GPR, defined as the percentage of points satisfying the condition that

Table 1: Quantitative results for the different evaluated models.

| Model | SSIM (%) | L1 | GPR (%) | #Parameters |
|---|---|---|---|---|
| 3D BiONet (input: noisy input sequence) | **91.1±2.2** | $1.02e^{-1} \pm 2.2e^{-2}$ | 50.5±4.8 | 178 M |
| 3D BiONet (input: noisy $5e^9$ particles) | $90.4 \pm 3.4$ | **$7.22e^{-2}\pm1.1e^{-2}$** | 71.9±5.1 | 178 M |
| Proposed Stacked 3D ConvLSTMs | $86.4 \pm 5.2$ | $1.04e^{-1} \pm 2.2e^{-2}$ | **83.8±3.4** | **5 M** |

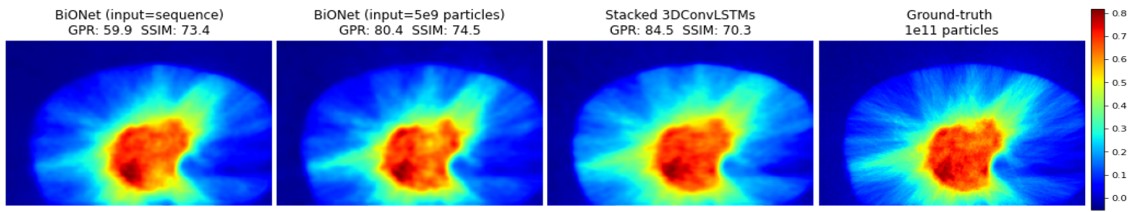

Figure 2: Visual comparison of denoised volumes by each model and the ground truth.

the gamma index (Low et al., 1998) is inferior to 1, is computed with a dose to agreement and tolerance on dose values of 3%/3mm within 30% - 100% of maximum dose. Figure 2 displays slices of denoised volumes by each model for further qualitative comparison. Unfortunately, we observe that both models give visually smoothed predictions compared to the fine-grained ground-truth. Both models remain competitive in terms of SSIM and L1 losses. Despite having the lowest number of parameters, our model outperforms BiONet with respect to the clinical assessment metric GPR. Future steps include ablation studies, establishing how noisy the training data can be without impacting performances and exploring further the potential of recurrent architectures for Monte-Carlo dose simulation denoising.

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
