# OpenReview forum: "Weakly supervised 3D ConvLSTMs for high precision Monte-Carlo radiotherapy dose simulations"
_MIDL.io/2021/Conference/Short — MIDL 2021 Poster_

### Official Review · Reviewer_86Pv · 2021-04-29

**Confidence:** 4
**Final Rating:** 3

**Summary:**

This paper proposes the use of a convolutional LSTM model, trained using the noise2noise paradigm, to denoise dose simulations to be used for radiotherapy planning. The motivation is that obtaining high precision dose simulations is time-consuming, so that it would be beneficial to instead use denoised versions of lower precision simulations. The authors validate their method against a benchmark model and show that it performs better, with less parameters, on a clinical assessment metric (the gamma passing rate).

**Strengths:**

Investigating faster methods for acquiring high-precision dose maps is important for radiotherapy planning; in particular for enabling closer to real-time simulation. The strength of this paper is that it shows that this can be done with a fairly light-weight CNN architecture, retaining a good clinical assessment metric, where the method additionally outperforms a more complex CNN model. The paper reads well and is structured in a good way.

**Weaknesses:**

My main concern with this paper is that the authors do not properly validate their main contribution: use of a  ConvLSTM modelling. They claim that ‘the model should learn the causal relationship that links the number of simulated particles and the level of precision of the resulting dose map.’ This should then have been validated experimentally; for example, by simply training the same model on only the 5e9 particle dose simulations (i.e., no RNN), to see if having the LSTM adds value in terms of the validation metrics.

**Deanonymize Review:**

no

**Detailed Comments:**

* Understanding the GPR metric seems, to me, non-trivial. I think it would help if you pointed to a good reference that explains it; especially since it describes your strongest result.
* In the results paragraph, change l1 metric to l1 loss.
* Please specify the size of your patches.
* I would try to use a slightly larger number of test images, at least 10, as it could improve the credibility of your results.
* Was the BioNet trained on the same patch size as the authors proposed model, or on whole images?

**Justification Of The Rating:**

It is a well written paper on an clinically impactful application, which readers might find interesting. Although I think that they could have better validated their contribution, I think it could be an interesting addition to MIDL’s short paper track, especially as many deep learning practitioners find exploring novel architectures valuable.

**Paper Type:**

both

**Special Issue:**

no

---

### Official Review · Reviewer_ZaJH · 2021-04-30

**Confidence:** 3
**Final Rating:** 3

**Summary:**

Radiotherapy treatment requires safe dose estimation, which is typically simulated using the monte-carlo technique. These simulations usually entail significant computational resources and time. The authors conjecture that performing MC simulations with fewer particles would require fewer computational resources which would be advantageous. By leveraging the *noise2noise* paradigm which only requires noisy data for training, one could denoise the noisy dose simulations to obtain equivalent results as a simulation with more particles.

The key contributions of the authors include (i) investigating the weakly supervised approach for denoising MC dose simulations, (ii) proposing the 3d convolutional-recurrent architecture for this task (iii) demonstrating comparable results of their approach with respect to one baseline method

**Strengths:**

- The authors present a novel solution for denoising radiotherapy dose simulations which uses only noisy dose simulations during the training.
- For benefitting from the volumetric and temporal nature of the simulations, the authors propose and employ a 3d recurrent and convolutional model.
- All results and comparison to baseline method were provided with the variance over 5 independent runs.

**Weaknesses:**

- A discussion on why the evaluation results seem better for the GPR metric vis a vis the L1 and SSIM metric, is not provided.
- The weighting of the L1 and SSIM during training has not been included.





**Deanonymize Review:**

yes

**Detailed Comments:**

- All used abbreviations could be expanded once before being used subsequent in the draft. (for example, VMAT, IMRT)
- Please use $mm^{3}$ when defining the resolution of the dose volume
 - An ablation study where the same N2N paradigm is used, but on a 3d convolutional architecture (without any recurrent connections) could be useful to understand the contribution of the choice of the architecture towards the final evaluation.
- (Optional and please skip if not practically possible) It would be interesting to see the supervised result before proceeding to the self-supervised result (i.e. can a network learn to reconstruct a high particle MC simulation given a low particle MC simulation if such paired data is provided during training).




**Justification Of The Rating:**

I think the proposed approach has significant merit.  With the provision of open source code and by including a few more validations/ablation studies, this work has potential to easily improve further.

**Paper Type:**

both

**Special Issue:**

no

---

### Meta-Review · Area_Chair_uywh · 2021-05-07

**Recommendation:** Accept (Poster)
**Confidence:** 5

**Metareview:**

The reviewers agree that the paper is interesting and should be accepted.

---

### Decision · Program_Chairs · 2021-05-11

Accept (Poster)